# Image and Text: Fighting the Same Battle?
# Super-resolution Learning for Imbalanced Text Classification

**Romain Meunier[1]** and **Farah Benamara[1,2]** and **Véronique Moriceau[1]** and **Patricia Stolf[1]**

(1) IRIT, Université de Toulouse, *CNRS, Toulouse INP, UT3*, Toulouse, France
(2) IPAL, CNRS-NUS-ASTAR, Singapore
`firstname.lastname@irit.fr`

## Abstract

In this paper, we propose SRL4NLP, a new approach for data augmentation by drawing an analogy between image and text processing: Super-resolution learning. This method is based on using high-resolution images to overcome the problem of low resolution images. While this technique is a common usage in image processing when images have a low resolution or are too noisy, it has never been used in NLP. We therefore propose the first adaptation of this method for text classification and evaluate its effectiveness on urgency detection from tweets posted in crisis situations, a very challenging task where messages are scarce and highly imbalanced. We show that this strategy is efficient when compared to competitive state-of-the-art data augmentation techniques on several benchmarks datasets in two languages.

## 1 Introduction

The performances of NLP classification tasks are largely influenced by the availability of good quality training data with sufficient number of representative instances. This is particularity salient for (highly) imbalanced text classification problems, like malware detection (Oak et al., 2019), relation extraction (Gao et al., 2019), hate speech detection (Schmidt and Wiegand, 2017) as well as urgency detection (Sánchez et al., 2023).

To augment minority classes, many data augmentation strategies have been proposed while ensuring models generalization. Roughly, these strategies can operate either at the data or model level. Data-based approaches generate synthetic new instances relying on lexical/syntactic variations (paraphrasing, substitution, etc.) or pre-trained generative models (e.g., GPT2, GPT3) (Anaby-Tavor et al., 2020; Yoo et al., 2021; Chia et al., 2022). Model-based approaches on the other hand fight class imbalance during training via (a) some dedicated loss functions by assigning higher wrong classification costs to classes with small proportion (Lin et al., 2020; Li et al., 2020), (b) specific architectures such as multi-task learning leveraging auxiliary tasks (Spangher et al., 2021) and knowledge distillation that trains smaller model with soft labels (Tan et al., 2022), or (c) text interpolation that mixes up original and perturbed samples in the hidden space (Chen et al., 2022). While data-based techniques are easy to implement with low computational cost compared to model-based ones (Wei and Zou, 2019), current methods have difficulty to generate meaningful instances as augmented data are often too close to the original ones, or semantically different which may result in instances that do not make sense to humans (Feng et al., 2021; Kamalloo et al., 2022). Prompt-based augmentation alleviates these issues but at the price of training large-scale language models. In addition, they turn to be sensitive to prompt design in terms of the choice and order of the input examples (Reynolds and McDonell, 2021).

In this paper, we propose a new approach for data augmentation by drawing an analogy between image and text processing: *Super-resolution learning* (Yang et al., 2019). This method is based on using high-resolution images to overcome the problem of low resolution images and has been successfully employed in image processing when images have a low resolution or are too noisy (Peng et al., 2016; Koziarski and Cyganek, 2018; Tian et al., 2023; Fukami et al., 2023). It has however never been adapted to NLP classification tasks. In this context, we make a parallel between image and text by considering noisy and short user generated content (e.g., social media posts) as a low resolution image and long well written documents (e.g., news papers, Wikipedia, scientific articles) as a high resolution image which enables augmentation with more diverse and high quality data with less cost. Mixed data from both types of resolution are then used to train supervised classifiers.

We measure the effectiveness of super-resolution data augmentation on single label text classification tasks which: (i) show a step-imbalanced distribution with a high fraction of minority classes (Buda et al., 2018), (ii) are characterized by a long-tailed distribution with many instances for a small number of classes, but few for the remaining classes (Henning et al., 2023), (iii) involve low-quality instances (e.g., few tokens, ill-formed with grammar spelling/syntactic issues), and (iv) are about media events such as disasters (e.g., flood, earthquakes), health (e.g., vaccines, cancer, suicide) and sport, that are more likely to be discussed in factual external sources of information, adding therefore meaningful context. Among NLP applications meeting these criteria (e.g., cybersecurity (Oak et al., 2019), medecine (Mayer et al., 2021; Ansari et al., 2021)), we experiment with urgency detection from tweets (Castillo, 2016; Alam et al., 2021), a very challenging task aiming at classifying messages posted during crisis situations according to fine-grained intention to act categories (such as human/infrastructure damages, warning/advice). In this task, messages that require an urgent and rapid action from rescue teams are scarce and highly imbalanced compared to non-urgent ones (Reuter et al., 2018; Sarioglu Kayi et al., 2020), making it a very interesting case study for super-resolution learning. In this paper, our contributions are:

- SRL4NLP, a simple yet effective adaptation of a well-established data augmentation strategy in image processing to text classification which does not rely on artificially generated data but on existing high quality texts.

- A study of the performances of this strategy on disaster management from social media relying on benchmark datasets in both English and French.

- A quantitative and qualitative evaluation in an out-of-type configuration where classifiers are evaluated in unseen event types during training. Our results suggest that super-resolution learning is achieving competitive results when compared to state-of-the-art data augmentation techniques.

In the following, Section 2 presents the original machine-learned super-resolution method and how we adapted it for NLP. Section 3 presents the datasets and Section 4 presents the experimental settings used in this study. Section 5 presents the results. We compare our approach to related work in Section 6. Finally, we conclude drawing some perspectives for future work.

## 2 Data Augmentation with Super-Resolution

### 2.1 Super-resolution for Image Processing

Data augmentation has first been explored in the field of image processing with algorithms for geometric transformations, color changes, filters, random erasing, etc. (see (Shorten and Khoshgoftaar, 2019) for a survey). Classifiers are mainly trained on high-resolution labeled data but do not perform well when tested on low-resolution data. Thus, super-resolution method, consisting in enhancing the resolution of an image, is largely approved and applies in many fields such as face recognition (Zou and Yuen, 2012), microscopic image (Nehme et al., 2018), the domains of flow (Liu et al., 2020; Yousif et al., 2021; Fukami et al., 2023) or medicine (van Sloun et al., 2019).

Super-resolution gained interest for deep learning with Peng et al. (2016) who evaluated different training strategies for CNN models to improve fine-grained category classification of very low resolution images. They artificially reduce the resolution of some high-resolution training data and show that mixing them during the training phase improves accuracy when models are tested on low-resolution images compared to training on either high or low-resolution only.

Thus, multi-resolution training, which randomly resizes images to different resolutions, allows to accommodate varying resolutions during testing and is used mainly to train CNN models. Recently, Tian et al. (2023) proposed ResFormer, a vision transformer trained on multi-resolution images (several scalings of images) which obtains better results than a transformer trained on single resolution.

### 2.2 SRL4NLP: Super-resolution for NLP

Inspired by these results, we draw an analogy between images and texts: (1) we consider that tweets are low-resolution texts as they are short and often low-quality, noisy texts; and (2) we propose to augment low-resolution training data with similar texts with a higher resolution (see Figure 1). We consider high-resolution texts as clean texts with a higher number of tokens than tweets. Instead of generating artificial high-resolution texts from low-

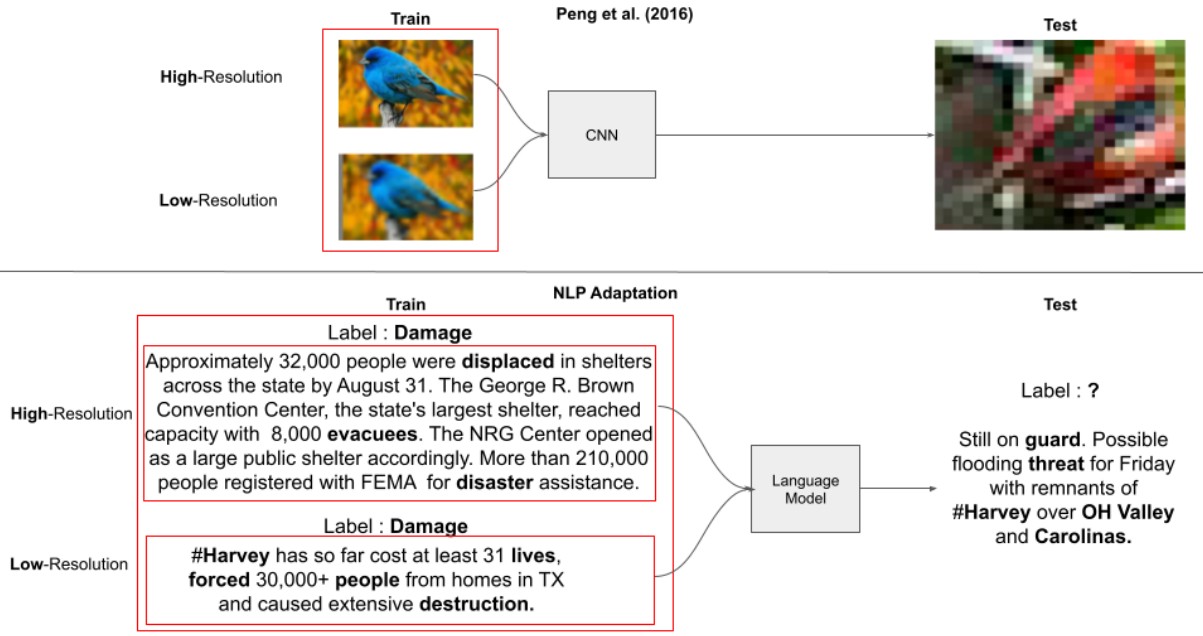

Figure 1: Adaptation of Peng et al. (2016) super-resolution learning to text classification, illustrated here on the task of urgency detection, our case study.

resolution ones before or during training as often done in image processing and to avoid additional training costs, we propose to augment data with existing texts such as newspaper, encyclopedic articles, or scientific publications, and to mix them with tweets for the training step.

SRL4NLP implies to have a high-resolution dataset about the same topics as the low-resolution one, and annotated with the same labels. If not available, we propose a method in Section 3 to collect and automatically annotate high-resolution data from structured texts, such as Wikipedia articles.

## 3 Datasets

The literature on emergency detection has been growing fast in the recent years and several datasets (mainly tweets) have been proposed to account for crisis-related phenomena.[1] Messages are annotated according to relevant categories that are deemed to fit the information needs of various stakeholders like humanitarian organizations, local police and firefighters. Well-known dimensions include relatedness to identify whether the message content is useful (Jensen, 2012) and situation awareness (also known as urgency) to filter out on-topic relevant vs. on-topic irrelevant information (Imran

et al., 2013; McCreadie et al., 2019; Sarioglu Kayi et al., 2020; Kozlowski et al., 2020). We evaluate several data augmentation strategies on different datasets in English and French since datasets for crisis management are available in these languages (see Table 1).

### 3.1 Low-Resolution datasets

- **KOZLOWSKI**$_{merge}$: This dataset is the largest French dataset for crisis management composed of 16,202 tweets about various expected (e.g., storms) and sudden crisis (e.g., explosions) events (Kozlowski et al., 2020; Bourgon et al., 2022). It contains tweets annotated for three urgency categories as well as 6 intentions to act categories: (1) URGENT that applies to messages mentioning HUMAN/MATERIAL DAMAGES as well as security instructions (ADVICE-WARNING) to limit these damages during crisis events, (2) NOT URGENT that groups SUPPORT messages to the victims, CRITICS or any OTHER messages that do not have an immediate impact on actionability but contribute in raising situational awareness, and finally (3) NOT USEFUL for messages that are not related to the targeted crisis. To be comparable with the following English dataset, some intent classes have been merged into more coarse-grained classes: HUMAN-DAMAGE and MATERIAL-DAMAGE into DAMAGE; SUPPORT and CRITICS into SOCIAL while

---

[1]See https://crisisnlp.qcri.org/ for an overview.

| | URGENT | | NOT URGENT | | NOT USEFUL | TOTAL |
|---|---|---|---|---|---|---|
| | DAMAGE | ADVICE_WARNING | SOCIAL | OTHER | | |
| KOZLOWSKI$_{merge}$ | 908 | 1,456 | 1,128 | 1,352 | 11,358 | **16,202** |
| HUMAID$_{merge}$ | 908 | 1,456 | 1,128 | 1,352 | 11,358 | **16,202** |
| FR$_{AUTO}$ | 603 | 276 | 153 | 0 | 0 | **1,032** |
| FR$_{AUTO\_NO\_SOCIAL}$ | 603 | 276 | 0 | 0 | 0 | **879** |
| EN$_{AUTO\_NO\_SOCIAL}$ | 1,160 | 0 | 0 | 0 | 0 | **1,160** |

Table 1: Distribution of each dataset in terms of urgency (3 classes) and intent (5 classes).

keeping imbalanced distribution with 71%, 8,98% and 5.60% for NOT USEFUL, SOCIAL and DAMAGE respectively.

- **HUMAID$_{merge}$**: It is an English dataset of 77,196 tweets annotated for crisis management following 10 intent categories and one NOT USEFUL class (Alam et al., 2021). This latter class having few instances (1,986) and as we wanted it to be a majority class as in KOZLOWSKI$_{merge}$, we added 9,372 NOT USEFUL instances from other crisis datasets annotated for relatedness (CRISISLEXT6 (Olteanu et al., 2014) and CRISISLEXT26 (Olteanu et al., 2015)). To allow a better comparison, we aligned HUMAID annotations with KOZLOWSKI$_{merge}$ by merging similar categories (e.g., injured/dead people and missing or found people into the unified label DAMAGE) while discarding irrelevant ones (e.g., no decision). We have also randomly selected the same number of tweets from the same classes as in KOZLOWSKI$_{merge}$ to have an equivalent distribution and a comparable size.

### 3.2 High-Resolution datasets

SRL4NLP implies to have a high-resolution dataset about the same topics as the low-resolution one, and annotated with the same labels, so that both datasets can be mixed during the training step. However, as there is no available high-resolution dataset for crisis management which fit our needs, we defined a simple but generic method to collect and annotate automatically high-resolution texts from existing structured documents. This method avoids additional computational costs due to artificial data generation and manual annotation. The selection and annotation of high-resolution texts from structured documents follow Algorithm 1, which annotates paragraphs of relevant documents according to the title of the section they belong to; assuming a set of labels $C$ of a classification task for a given use case and a set of documents $D$ related to this use case structured in terms of sections $S$ and paragraphs $P$, such that:

- $C = \{(c_1, k_1), ..., (c_n, k_n)\}$ where each label $c_i$ is triggered by a set of keywords $K = \{k_1, ..., k_l\}$, and

- $D = \{d_1, ..., d_m\}$ where $d_j = <(S_j, P_i^+)^+ >$ such that $S_j = \{s_{j,1}, ..., s_{j,n}\}$ the $n$ sections of $d_j \in D$, and $P_n = \{p_{n,1}, ..., p_{n,r}\}$ the $r$ paragraphs of $S_j$.

---

**Algorithm 1:** HIGH-RESOLUTION TEXT SCRAPPER AND ANNOTATION

**Input:** $C, K, D, S, P$
**Output:** label($p_r$)

1 **for** $d_i$ in $D$ **do**
2    **for** $s_{i,j}$ in $S_i$ **do**
3      **for** $c_n, k_n$ in $C$ **do**
4        **if** $k_n$ in $s_j$ **then**
5          **for** $p_r$ in $P_j$ **do**
6            label($p_r$) = $c_n$

---

For our use case, we decided to use Wikipedia as external source as it contains a large variety of factual and valuable information for understanding the nature, causes, impacts, and responses to different crises (e.g., damage and weather reports, responses and mitigation efforts, media coverage and public reactions) making it well suited for super-resolution. In addition, it is copyright free, easy to scrap, and comes with a structured format with dedicated sections and paragraphs for each Wikipedia page. Thus, the following datasets, built with Algorithm 1, are composed of high-resolution paragraphs from Wikipedia, used to augment all minority classes, except NOT USEFUL, the majority class:

- **FR$_{AUTO}$**: the set of relevant documents $D$ is composed of the 152 links in the French Wikipedia page *Natural disasters by year*[2]. The sets of keywords $K$ used for the matching of section titles

---

[2] https://fr.wikipedia.org/wiki/Cat%C3%A9gorie:Catastrophe_naturelle_par_ann%C3%A9e

with classes are given in Appendix A. This results in 1,032 labelled paragraphs, each of them being related to a crisis type given by the document title. Among them, we manually checked 144 randomly selected paragraphs and agreed with the automatic annotation (see an example in Appendix C).

- **FR**$_{AUTO\_NO\_SOCIAL}$: unlike Twitter, texts in Wikipedia which can be labelled as SOCIAL (i.e. critics, support) are rare. For this reason, we have created a subset of FR$_{AUTO}$, named FR$_{AUTO\_NO\_SOCIAL}$, which contains only the data labelled DAMAGE and ADVICE_WARNING.

- **EN**$_{AUTO\_NO\_SOCIAL}$: in the same way, 213 links in the English Wikipedia page *Natural disasters by year*[3] have been scrapped leading to the automatic selection and annotation of 1,160 paragraphs as DAMAGE or ADVICE_WARNING (see Appendix B for the English matching lexicon).

Table 1 shows the distribution of these augmented datasets.

## 4 Models and Evaluation Protocol

### 4.1 Experimental Settings

For testing the datasets in French, we used **FlauBERT**$_{Fine-Tuned}$ which is the best performing model for intent classification on the French dataset according to (Kozlowski et al., 2020). It is a FlauBERT model (Le et al., 2020) that has been fine-tuned on 358,834 unlabelled tweets posted during crises. For English, we used **RoBERTa** (Liu et al., 2019) which is reported to be efficient for crisis management in English (Koshy and Elango, 2022; Rocca et al., 2023; Madichetty et al., 2023). For both models, we used the Focal Loss with an Adam optimizer.

Regarding the classification task, we do not report any comparative analysis with existing mainstream large language model such as GPT for three main reasons. First, our aim is to design a simple approach that requires very low computational cost. Second, as we used GPT3 to generate the new augmented data and so learned from these data, we decided to not use it for classification because results may be biased. Finally, we also experimented classification with GPT3 on 100 tweets of KOZLOWSKI$_{merge}$. However, it was unable to distinguish OTHER and NOT USEFUL messages, achieving therefore low performances compared

to FlauBERT (e.g. GPT3 was only able to correctly classify 35% of messages into intention to act categories).

Following the general trends in crisis management (Kersten et al., 2019; Algiriyage et al., 2021; Bourgon et al., 2022), we designed an *out-of-type* evaluation protocol by training on a pool of events related to different types of crises (e.g., Hurricane, Storm) and testing on a particular different type (e.g., Earthquake). The aim is to evaluate if a model can deal with new types of crisis, which is crucial to ensure the portability of the models to unseen events.

Each corpus contains different types of crisis: Fire, Flood, Storm, Hurricane, Collapse and Terrorist attack for KOZLOWSKI$_{merge}$ and Fire, Flood, Storm, Hurricane, Earthquake for HUMAID$_{merge}$. Thus for each dataset, an experiment consists in the average of $n$ runs, each run with $n-1$ crisis types for training and the remaining crisis type for testing. For HUMAID$_{merge}$, as a random data sampling is done so that the dataset can be comparable with the distribution of the French one and as the initial English dataset is pretty large (around 86K), we created 5 independent batches of 16,202 random tweets per batch conserving the same class distribution as KOZLOWSKI$_{merge}$ and reported the averaged macro F-scores.

For both datasets, we replaced all numbers and URLs by a dedicated token and removed user mentions.

### 4.2 Methods

In addition to our super-resolution data augmentation method, we evaluate several state-of-the-art approaches:

#### 4.2.1 Data-based approaches

- **Synonym replacement** (Zhang et al., 2015): This method substitutes a random word in a text by one of his synonym selected in a thesaurus based on a language model prediction. We used the implementation from the NLPAUG library.[4]

- **Contextual augmentation**: Based on Kobayashi (2018), this method requires a deep learning model calculating the word probability at a given position based on its context. Then, it replaces the word at this position by the predicted word.

- **Back Translation** (Yu et al., 2018): This method translates a text from a source language

---

[3]https://en.wikipedia.org/wiki/Category:Natural_disasters_by_year

[4]https://github.com/makcedward/nlpaug

| Text | Method |
|------|--------|
| (1) #Harvey has so far cost at least 31 lives, forced 30,000+ people from homes in TX and caused extensive destruction. | Original tweet |
| (2) #Harvey has cost at least 31 lives so far, forced more than 30,000 people to leave their homes in TX and caused considerable destruction. | Back translation |
| (3) # Moreve It has so far cost at quigo 31 liveds, forced 3 et+ over from homes in TX that causes some extensive destruction. | Synonym replacement |
| (4) Old #Harovey the has so was far we cost at high leostast 1 31 lives, forced 30,000+ people from homes in TX and caused so extensive place destruction. | Contextual augmentation |
| (5) The impact of #Harvey has been devastating, with 31 lives lost, over 30,000 people displaced from their homes in TX, and widespread destruction. Let's stand together and support those affected by this crisis. #SupportForHarveyVictims #RebuildingTogether | Generative augmentation |
| (6) Approximately 32,000 people were displaced in shelters across the state by August 31. The George R. Brown Convention Center, the state's largest shelter, reached capacity with 8,000 evacuees. The NRG Center opened as a large public shelter accordingly. More than 210,000 people registered with FEMA for disaster assistance. | **Super-resolution** |

Table 2: Examples of augmented data from one tweet.

into a target language and then translates it back to the source. We used Helsinki-NLP/OPUS-MT-train (Tiedemann and Thottingal, 2020)[5] to translate the French dataset into English and then back into French. For the English dataset, we translated it into French and then back into English.

- **Generative augmentation**: Inspired by Yoo et al. (2021), we used GPT3 (Brown et al., 2020) to generate new data similar to tweets of our datasets. The prompt asks to generate 10 new tweets similar to 5 tweets from the original dataset that share the same label but can be from different crises.

To be able to make comparison between classification models, we used the above methods to augment data with the same distribution as $FR_{AUTO\_NO\_SOCIAL}$ to augment $KOZLOWSKI_{merge}$ and $EN_{AUTO\_NO\_SOCIAL}$ to augment $HUMAID_{merge}$. Table 2 presents examples of augmented data obtained from one tweet with the above methods, showing for some of them the low quality of what is generated.

### 4.2.2 Model-based approaches

We also compare with the following two model-based methods as they have been shown to be quite effective for crisis management.

- **Manifold Mixup** (Verma et al., 2019): It has been adapted to crisis management by Chowdhury et al. (2020a). It makes semantic interpolation during training, obtaining neural networks (here, $FlauBERT_{Fine-Tuned}$ for French and RoBERTa for English) with smoother decision boundaries.[6]

- **Multi-task learning** (Ye et al., 2019): This strategy trains several layers on different but related tasks to improve the performance. This learning architecture has already been used for crisis management by Kozlowski et al. (2020) and Khanal and Caragea (2021). Following Kozlowski et al. (2020), the tasks are utility (useful vs. not useful), urgency (urgent vs. not urgent vs. not useful) and intention (5 classes) detection, the latter being our main task. For all tasks, we used the Focal loss function and $FlauBERT_{Fine-Tuned}$ for French and **RoBERTa** for English.

## 5 Results

### 5.1 Quantitative results

Table 3 presents the results obtained by $FlauBERT_{Fine-Tuned}$ on the $KOZLOWSKI_{merge}$ dataset (5 classes) and its augmented versions. The model obtains the best results on the dataset augmented with $\mathbf{FR}_{AUTO\_NO\_SOCIAL}$ via the automatic super-resolution method. We note that augmenting only the classes DAMAGE and ADVICE-WARNING with few high-resolution texts from Wikipedia can even help improving the performance on the class SOCIAL which has not been augmented. Indeed, Wikipedia contains informative data, similar to what is found in tweets for both classes DAMAGE and ADVICE-WARNING whereas it contains almost no SOCIAL data. Thus, augmenting only the first two classes helps the model to be more discriminant on the class SOCIAL.

Table 4 presents the results obtained by RoBERTa on the $HUMAID_{merge}$ dataset (5 classes)

---

[5]https://github.com/Helsinki-NLP/OPUS-MT-train

[6]https://github.com/vikasverma1077/manifold_mixup

| Dataset | | Damage | Advice_Warning | Social | Other | Not Useful | Total |
|---|---|---|---|---|---|---|---|
| KOZLOWSKI$_{merge}$ | | 54.87 | 46.94 | 54.62 | 23.86 | 82.83 | 52.62 |
| Manifold Mixup | | 55.95 | 46.67 | 50.65 | 26.00 | 82.64 | 52.38 |
| Multi-task | | 57.76 | 47.15 | 50.99 | **30.50** | 82.86 | 53.85 |
| KOZLOWSKI$_{merge}$ | +Back Translation | 48.94 | 46.48 | 49.25 | 26.40 | **83.00** | 50.81 |
| KOZLOWSKI$_{merge}$ | +Generative augmentation | 52.77 | 47.66 | 50.74 | 22.09 | 81.83 | 51.02 |
| KOZLOWSKI$_{merge}$ | +Contextual augmentation | 58.37 | 48.04 | 50.84 | 24.41 | 82.05 | 52.74 |
| KOZLOWSKI$_{merge}$ | +Synonym replacement | 57.04 | 47.38 | 51.35 | **28.12** | 81.22 | 53.02 |
| FR$_{AUTO}$ | | 55.67 | 47.75 | 50.18 | 22.51 | 81.37 | 51.50 |
| FR$_{AUTO\_NO\_SOCIAL}$ | | **58.65** | **48.33** | **56.00** | 25.73 | 82.26 | **54.19** |

Table 3: Results of FlauBERT$_{Fine-Tuned}$ in terms of average F-score on KOZLOWSKI$_{merge}$ for *out-of-type* experiment.

| Dataset | | Damage | Advice_Warning | Social | Other | Not Useful | Total |
|---|---|---|---|---|---|---|---|
| HUMAID$_{merge}$ (5 batches) | | 69.87 | 73.04 | 82.51 | 39.18 | 92.05 | 71.33 |
| Manifold Mixup | | 70.42 | 72.78 | 82.30 | 41.26 | **92.26** | 71.81 |
| Multi-task | | 73.99 | 68.04 | 81.16 | 37.43 | 91.47 | 70.42 |
| HUMAID$_{merge}$ | +Back Translation | 74.86 | 72.66 | 81.21 | 37.31 | 90.77 | 71.36 |
| HUMAID$_{merge}$ | +Generative augmentation | 68.46 | 71.14 | 75.89 | 31.26 | 91.12 | 67.57 |
| HUMAID$_{merge}$ | +Contextual augmentation | 87.59 | 71.11 | 82.06 | 41.78 | 91.56 | 74.82 |
| HUMAID$_{merge}$ | +Synonym replacement | **87.61** | **73.50** | 81.17 | 41.03 | 91.78 | **75.02** |
| EN$_{AUTO\_NO\_SOCIAL}$ | | 84.90 | 73.02 | **83.52** | **42.03** | 91.38 | 74.97 |

Table 4: Results of RoBERTa in terms of average F-score on HUMAID$_{merge}$ for *out-of-type* experiment.

| Dataset | | Damage | Advice_Warning | Social | Other | Not Useful | Total |
|---|---|---|---|---|---|---|---|
| HUMAID$_{merge}$ (5 batches) | | 71.23 | 65.58 | 93.66 | 42.31 | 80.48 | 70.65 |
| Manifold Mixup | | 70.79 | 67.12 | 93.57 | 42.58 | 80.12 | 70.83 |
| Multi-task | | 73.39 | 68.49 | 78.63 | 40.35 | 93.87 | 70.95 |
| HUMAID$_{merge}$ | +Back Translation | 70.99 | 72.37 | 93.49 | 41.76 | 79.13 | 71.54 |
| HUMAID$_{merge}$ | +Generative augmentation | 69.77 | 64.63 | 93.88 | 33.68 | 72.05 | 66.80 |
| HUMAID$_{merge}$ | +Contextual augmentation | 69.33 | 86.84 | 93.44 | 45.35 | 80.00 | 74.99 |
| HUMAID$_{merge}$ | +Synonym replacement | **72.63** | 86.14 | 93.46 | 44.65 | 78.48 | 75.07 |
| EN$_{AUTO\_NO\_SOCIAL}$ | | 71.52 | **86.99** | **93.88** | **45.74** | **81.70** | **75.97** |

Table 5: Results of RoBERTa in terms of average F-score on HUMAID$_{merge}$ for *out-of-type* experiment without the test run on Earthquake.

and on its augmented versions. Except *Generative augmentation* and *Multi-task*, all augmentation methods outperform the baseline on the original dataset. We can make a similar observation as for the French dataset: augmenting only the class DAMAGE with EN$_{AUTO\_NO\_SOCIAL}$ helps improving the performance on the class SOCIAL. SRL4NLP is second behind *Synonym replacement* and its results are statistically significant compared to those obtained on the original HUMAID$_{merge}$ dataset (*p-value*=1.68e-12 with the McNemar test). When looking into details at the 5 runs on each crisis type, we noticed that Earthquake was an outlier for which our model underperformed. The test run on Earthquake has a F-score of 70.99 when augmented with SRL4NLP whereas the F-score is 74.05 on the original dataset: this crisis type has the strongest decrease compared to other cri-

sis types. This may be explained by the fact that only 57 paragraphs have been added to augment the data for earthquake, representing much less data compared to other crisis types (at least 100 paragraphs). Table 5 presents the results obtained without the test run on earthquake. In this configuration, SRL4NLP outperforms all state-of-the-art augmentation methods and is more efficient for all classes except DAMAGE.

Concerning recall and precision, FlauBERT$_{Fine-Tuned}$ obtained a precision of 55.24 (resp. 55.98) and a recall of 53.33 (resp. 55.53) on KOZLOWSKI$_{merge}$ (resp. when augmented with SRL4NLP). RoBERTa obtained a precision of 72.01 (resp. 74.03) and a recall of 73.11 (resp. 77.85) on HUMAID$_{merge}$ (resp. when augmented with SRL4NLP). SRL4NLP increases mainly recall, which fits particularly

with the needs of crisis management where the emergency services need to be aware of all urgent messages.

## 5.2 Error analysis

A manual error analysis highlights three main causes of misclassification. For example, in Table 6, the predicted label in messages (1) and (2) seems more accurate (support) than the gold label. Indeed, the tweet (1) is about displaced people so it can be considered as DAMAGE but the main objective of the tweet is an intent to support these people. In tweet (2), the support is represented by the hashtag *#PrayForGreece*. In tweet (3), the word *damage* misled the model to consider this tweet as DAMAGE. Finally, message (4) could have multiple labels: the gold label ADVICE-WARNING since the tweet declares a state of emergency but also the predicted one DAMAGE since it mentions people evacuation.

## 6 Related Work

### 6.1 Data Augmentation in NLP

Various methods have been developed to generate artificial data in order to enhance the size and the quality of training datasets, avoiding collecting and annotating data manually (Feng et al., 2021). Language models can be used for back translation (Yu et al., 2018) or for synonym replacement and contextual insertion (Zhang et al., 2015; Kobayashi, 2018), keeping the semantic of the original sentences in the artificial data. Coulombe (2018) proposed error injection to be more robust to the noise found in social media posts. Rule-based methods can also be used to add, replace or swap words in a sentence (Wei and Zou, 2019), avoiding using costly language models. Recently, generative language models such as GPT3 have also been employed to generate data and augment existing datasets in a few/zero-shot scenario by using the class label and dedicated prompts as cue for the model (Anaby-Tavor et al., 2020; Yoo et al., 2021; Chia et al., 2022; Ocampo et al., 2023).

Model-based approaches on the other hand make use of multi-task learning leveraging auxiliary tasks (Spangher et al., 2021), knowledge distillation that trains smaller model with soft labels (Tan et al., 2022), and text interpolation that mixes up original and perturbed samples in the hidden space (Chen et al., 2022).

In this paper, we relied on advances in data augmentation for image classification and propose SRL4NLP, a novel augmentation paradigm adapting the well-established super-resolution learning to text classification. Our approach allows for adding new linguistic patterns in the train set relying on structured external textual sources and avoids generating any new data by using existing high-resolution data for a low computational cost.

### 6.2 NLP for Crisis Management

Social media, such as Twitter, can give precious real-time information in crisis situations (Reuter et al., 2018) (e.g., more than 1,000,000 tweets posted during in Türkiye and Syria earthquake in 2023 (Toraman et al., 2023)), making NLP-based crisis management a hot research topic. Recently, shared tasks have proposed, such TREC-IS (Incident Streams track)[7] (McCreadie et al., 2019, 2020), and CrisisFACTS2022[8] (McCreadie and Buntain, 2023) which allows the development of classifiers in three settings: (1) relatedness (Is a message useful/relevant for emergency departments?), (2) urgency (Is a message urgent for emergency departments? Possibly what is the urgency degree?), and (3) intents (damage reports, warnings, critics, etc.).

Classifiers are trained in a supervised way with either traditional feature-based learning algorithms (Alam et al., 2021; Li et al., 2018; Kaufhold et al., 2020) or deep learning architectures (Caragea et al., 2016; Neppalli et al., 2018; Kersten et al., 2019; Kozlowski et al., 2020; Chowdhury et al., 2020b; Liu et al., 2021; Wang et al., 2021; Dusart et al., 2021). Overall, results show that intent detection is the most challenging task due to the extremely imbalanced nature of crisis datasets: urgent messages are minority (e.g., about 3.87% and 1.93% for infrastructure/human damages respectively in CrisisFACTS).

The imbalanced issue has however received less attention in the literature. Among existing works, Bayer et al. (2021) used text generation, Kruspe et al. (2018) relied on automatic round-trip translation augmenting under-represented classes in the TREC-IS dataset. Ramachandran and Ramasubramanian (2021) proposed a synonym and hypernym replacement strategy based on WordNet. Finally Chowdhury et al. (2020a) explored text interpolation via Manifold Mixup in a multilingual setting

---

[7] https://www.dcs.gla.ac.uk/~richardm/TREC_IS/
[8] https://crisisfacts.github.io/

| Text | Gold label | Predicted Label |
|---|---|---|
| (1) Want to help support the thousands of residents displaced by the FortMcMurray massive wildfire Don t get scammed | Damage | Social |
| (2) Athens is either under wildfires or under storms i hate this summer PrayForGreece | Not Useful | Social |
| (3) Fort McMurray Trudeau arrives to view wildfire damage | Other | Damage |
| (4) Alberta Canada declared a state of emergency after a wildfire forced more than numero numero people to evacuate | Advice_Warning | Damage |

Table 6: Examples of misclassification by RoBERTa on $\text{HUMAID}_{merge}$ dataset.

with BERT. The use of super-resolution in the field is novel.

## 7 Conclusion

By drawing an analogy between image and text processing, we have proposed SRL4NLP, a new approach based on super-resolution learning to augment low-resolution imbalanced training data (for example, tweets) with similar texts having a higher resolution (paragraphs).

We have evaluated its effectiveness on urgency detection from tweets posted in crisis situations, relying on imbalanced benchmark datasets in both English and French. We have compared the classification performances over these datasets augmented with few data obtained with several data augmentation methods. The results show statistically significant F-scores in out-of-type evaluation setting, outperforming existing data-based and model-based augmentation approaches. SRL4NLP has also a very low computational cost since it avoids the generation of artificial data and their manual annotation.

For future work, we plan to rely on other external sources for augmenting our data (e.g., news, government reports, return of operating experiences) as well as experiment with other use cases (see below).

## Acknowledgements

The research of Farah Benamara is partially supported by DesCartes: The National Research Foundation, Prime Minister's Office, Singapore under its Campus for Research Excellence and Technological Enterprise (CREATE) program.

## Limitations

In this paper, we have evaluated SRL4NLP on crisis management benchmark datasets in English and French but the method and the evaluation protocol for this use case can be extended to datasets in other languages since Wikipedia pages are available in many languages.

We think that it can be also easily transfered to other classification use cases for which low-resolution datasets and high-resolution comparable external resources are available: for example, hate speech detection by augmenting datasets of tweets with posts on discussion forums; political opinion detection by augmenting datasets from social media with political speeches; medical diagnosis by augmenting patient records with medical publications, etc.

## Ethics Statement

The data that was used for conducting the experiments is composed of texts from the public domain taken from either datasets publicly available to the research community, or Wikipedia. The datasets are anonymized and contain no offensive or abusive language. They were collected before Twitter changed to X and conform to the Twitter Developer Agreement and Policy that allows unlimited distribution of either the numeric identification number or the textual content of each tweet.

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

# A French Lexicon

**Damage**: Bilan, Victimes, Répercussion, Dommage, Impact, Conséquences, Dégâts, Infrastructure, Humain, Panne, Séquelle, Pertes, Rescapés, Dévastation, Effet, Destruction

**Advice_Warning**: Précédant, Repère, Chronologie, Précurseur, Préparatifs, Contexte, Situation, Préparation, Prémice, Prévision, Évolution, Plan

**Social**: Réactions, Secours, Reconstruction, Presse, Communiqué, Judiciaire, Controverse, Complotiste, Instrumentalisation, Solidarité, Sociale, Aide, Restauration, Hommage, Commémoration, Communauté

# B English Lexicon

**Damage**: Effects, Fatalities, Impact ,Damage, Casualties, Outbreak, Destruction, Victim, Death, Result, Outcome, Rupture, Aftermath, Incident, Losses, Outages, Injuries, Trapped, Body

# C Scrapper and Automatic Annotation

# Hurricane Harvey

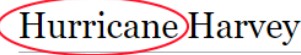

文A 32 languages ∨

## Meteorological history

(...)

## Preparations

(...)

## Impacts in the United States

The widespread and catastrophic effects of Hurricane Harvey resulted in one of the costliest natural disasters in United States history. An estimated 300,000 structures and 500,000 vehicles were damaged or destroyed in Texas alone. The storm also spawned 53 tornadoes across six states. The National Oceanic and Atmospheric Administration estimated total damage at $125 billion, with a 90% confidence interval of $90–160 billion. The scope of flooding in areas with low National Flood Insurance Program (NFIP) participation lends to the large uncertainty in the damage total. This ranks Harvey as the costliest tropical cyclone on record in the country alongside Hurricane Katrina in 2005. However, accounting for inflation and cost increases since 2005, the National Hurricane Center considers Harvey the second-costliest.[16] Harvey was the costliest natural disaster recorded in Texas at the time,[3] until it was surpassed in February 2021 by a severe winter storm that crippled the state's power grid, which was estimated to have cost at least $195 billion (2021 USD) in damages in Texas.[31]

Nationwide, 107 people died in storm-related incidents: 103 in Texas, 2 in Arkansas, 1 in Tennessee, and 1 in Kentucky. Of the deaths in Texas, 68 were from the direct effects of Harvey, the highest such number in the state since 1919.[16]

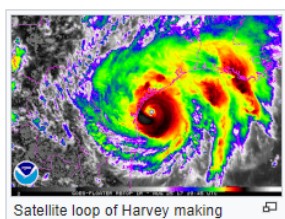

Satellite loop of Harvey making landfall in Texas on August 26

---

Lexicon for the label DAMAGE:

Effects, Fatalities, Impact, Damage, Casualties, Outbreak, Destruction, Victim, Death, Result, Outcome, Rupture, Aftermath, Incident, Losses, Outages, Injuries, Trapped, Body

Legend:

◯ (red) Crisis type in the page title

◯ (green) Word in the lexicon and in the section title

▭ (blue) Labelled high-resolution paragraph

Figure 2: Example of the automatic annotation of Wikipedia paragraphs (https://en.wikipedia.org/wiki/Hurricane_Harvey).