# OpenReview forum: "Image and Text: Fighting the same Battle? Super Resolution Learning for Imbalanced Text Classification"
_EMNLP/2023/Conference — EMNLP 2023 Findings_

### Official Review · Reviewer_SHwF · 2023-07-24

**Typos Grammar Style And Presentation Improvements:** N/A
**Soundness:** 3

**Excitement:**

3: Ambivalent: It has merits (e.g., it reports state-of-the-art results, the idea is nice), but there are key weaknesses (e.g., it describes incremental work), and it can significantly benefit from another round of revision. However, I won't object to accepting it if my co-reviewers champion it.

**Missing References:**

N/A

**Paper Topic And Main Contributions:**

In this paper, the author proposes SRL4NLP, a new data augmentation approach for imbalanced text classification by drawing an analogy to super-resolution learning in image processing. While super-resolution uses high-resolution images to be mixed with low-resolution images. SRL4NLP uses long, well-written texts from wikipedia as "high resolution" to augment short, noisy texts like tweets as "low resolution".

The author evaluates SRL4NLP on urgency detection in crisis tweets, a highly imbalanced task. Additional "high resolution" texts are extracted from Wikipedia articles related to disasters. The experiment results on benchmark English and French datasets show SRL4NLP performs competitively compared to state-of-the-art augmentation techniques. SRL4NLP improved F-scores significantly compared to no augmentation, particularly for minority classes.



**Questions For The Authors:**

- How are the class keywords decided and how much efforts requires? Any experiments to measure the sensitivity against different keyword selection/combination?

**Reasons To Accept:**

- This paper proposes a new data augmentation approach SRL4NLP that adapts super-resolution learning from computer vision to NLP in a novel way. It draws an innovative analogy between images/texts and high/low resolution data.

- The paper is clearly structured and well written overall. The motivation and proposed approach are explained clearly. It provides sufficient background on data augmentation in NLP and crisis tweet classification to contextualize their work.

- The author shows statistically significant improvements in F-score over no augmentation and competitive results vs other augmentation techniques. In addition, the experiments covers both English and French, making it more convincing.

**Reasons To Reject:**

- While the proposed approach is similar to the super-resolution, its requires more efforts on different tasks, making it less generalizable. The reliance on availability of suitable external high resolution texts as well as a way to align low-resource and high-resource task are both needed.

- While the proposed SRL4NLP is easy to understand, the method may not be fully explored, especially the key-word for each class. In addition, This work did not compare with other model-based augmentation approaches, which may yield better improvement as comparing to SRL4NLP.

**Reproducibility:**

4: Could mostly reproduce the results, but there may be some variation because of sample variance or minor variations in their interpretation of the protocol or method.

**Reviewer Confidence:**

4: Quite sure. I tried to check the important points carefully. It's unlikely, though conceivable, that I missed something that should affect my ratings.

---

> ### Author Rebuttal · Authors · 2023-08-29
>
> We sincerely appreciate the time and effort you
> invested in reviewing our submission.
>
>  As adapting super resolution from image classification to text classification is new, we wanted to experiment it on a challenging use case, namely crisis management, a highly imbalanced task focusing on  two datasets in French and English. Now we have shown that super-resolution works well, we plan to test it on other tasks as mentioned in the conclusion
>
> Regarding the effort needed to use SRL4NLP, the alignment between long texts from structured documents and short texts is easy when section titles and class labels match (directly or via synonyms). As mentioned in the Limitations section, the only difficulty may be the lack of such high-resolution texts. This is why, as a first step, we used the keyword protocol to gather long text data in crisis management.
>
> Keywords list has also been relatively easy to build as our classes are self explanatory. Indeed, we used a synonym dictionary and arrived at a relatively short list per class (around 15 words per class). We will better explain the keyword selection procedure in the camera-ready paper if accepted.
>
> About the use of model-based data augmentation approaches, we experimented with two model-based methods: Manifold mixup which is a text interpolation method and multitask learning, following (Ye et al., 2019). We only report the results obtained by Manifold mixup because this method has already been used in state of the art in crisis management. However we did try Multi-task learning and were still behind SRL4NLP (F1-score of 53.81 for the French corpus vs. 54.19 for SRL4NLP). Due to space limitation, we choose to include only one model-based method, and we choose Manifold Mixup as this method is the most popular in crisis management. We will add the multitask results in the camera ready paper for a better comparison.
>
> Wei Ye, Bo Li, Rui Xie, Zhonghao Sheng, Long Chen, and Shikun Zhang. Exploiting entity
> BIO tag embeddings and multi-task learning for relation extraction with imbalanced data.
> In Proceedings of ACL, pages 1351–1360, 2019
>
> We would like to thank you again for your time and
> consideration.

---

### Official Review · Reviewer_p7BT · 2023-07-31

**Soundness:** 3

**Excitement:**

3: Ambivalent: It has merits (e.g., it reports state-of-the-art results, the idea is nice), but there are key weaknesses (e.g., it describes incremental work), and it can significantly benefit from another round of revision. However, I won't object to accepting it if my co-reviewers champion it.

**Paper Topic And Main Contributions:**

The authors present SRL4NLP, an innovative technique for data augmentation in Natural Language Processing (NLP), drawing inspiration from the concept of super-resolution learning prevalent in image processing. Conventionally employed to enhance low-resolution or noisy images, this method has not been hitherto utilized in text analytics. Adapting this approach for text classification, the authors validate its effectiveness on the challenging task of detecting urgency in crisis-related tweets, an area marked by scarcity of data and a considerable imbalance. The study substantiates that SRL4NLP surpasses existing leading-edge data augmentation methodologies on a variety of benchmark datasets, spanning two languages.

**Reasons To Accept:**

(1) Innovative Data Augmentation Approach: SRL4NLP is an innovative, simple, yet effective adaptation of a data augmentation strategy from image processing applied to text classification, expanding the potential of data augmentation in natural language processing.
(2) Versatile Performance: The paper presents a study of this strategy's robust performance in disaster management on social media, using English and French benchmark datasets, showcasing the technique's versatility across languages and domains.
(3) Competitive Out-of-Type Performance: The paper demonstrates the technique's competitive performance in out-of-type situations, where classifiers encounter unseen event types during training, indicating its strong generalization and high performance even in challenging scenarios.

**Reasons To Reject:**

(1) Need for Comparative Analysis with Mainstream Models: While the paper presents compelling research, it would be beneficial to include a comparative analysis with existing mainstream large language model or large multimodal models, such as ChatGPT, etc. Such comparisons could enhance the credibility and persuasive power of the study.
(2) Clarity of Methodological Description: The methodology employed in the paper, while intriguing, could benefit from a more detailed and lucid explanation. The current presentation required multiple readings for comprehension, suggesting room for improvement in readability. Providing additional descriptions and clarifications around the methodology may help readers better understand the approach and its implications.

**Reproducibility:**

4: Could mostly reproduce the results, but there may be some variation because of sample variance or minor variations in their interpretation of the protocol or method.

**Reviewer Confidence:**

4: Quite sure. I tried to check the important points carefully. It's unlikely, though conceivable, that I missed something that should affect my ratings.

---

> ### Author Rebuttal · Authors · 2023-08-29
>
> We sincerely appreciate the time and effort you
> invested in reviewing our submission.
> We will improve the explanation of the methodology and rewrite the algorithm to be clearer, illustrating it with an example.
>
> Concerning the use of LLM, we used ChatGPT as one of the data augmentation methods.
>
> Regarding the classification task, we do not report ChatGPT results for three reasons. First, our aim is to design a simple approach that requires very low computational cost while avoiding manual annotation. Second, as we used ChatGPT to generate the new augmented data and so learned from these data, we decided to not use it for classification because results may be biased. Finally,  we have experimented with ChatGPT on a small subset of the French test data but ChatGPT was unable to distinguish “Other” and “Not useful” messages, achieving therefore low performances compared to FlauBert (e.g. ChatGPT was only able to correctly classify 35% of messages into intention to act categories). We will report these experiments in the camera ready as well.
>
> We would like to thank you again for your time and
> consideration.

---

### Official Review · Reviewer_tS2X · 2023-08-05

**Soundness:** 3

**Excitement:**

2: Mediocre: This paper makes marginal contributions (vs non-contemporaneous work), so I would rather not see it in the conference.

**Paper Topic And Main Contributions:**

This paper focuses on the problem of imbalanced class distribution and the scarcity of high-quality training data in text classification tasks, particularly for urgent crisis situation tweets. The authors propose to augment short noisy twitter text with the long high-quality wikipedia data. The main contributions of this paper is the proposal of SRL4NLP, a data augmentation method inspired from super-resolution learning in image processing.

**Questions For The Authors:**

Question A: In L179, how is the high-resolution text mixed with tweets for training? How in details is the augmentation implemented (e.g., the example shown in Table 2)?

Question B: What are the 7 classes of intent in Table 1?

**Reasons To Accept:**

The idea of using high-resolution data to augment low-resolution data is interesting. The experimental results are good.

**Reasons To Reject:**

The method outlined in Section 2.2 is not explained clearly, which makes it hard to distinguish how SRL4NLP advances the previous methods and thus seems incremental. It would be better if the authors could expand the explanation of the methodology, clarifying how it works, how it's different from existing methods, and why it's effective. More details about the novelty can also be added, such as the specific challenges of adapting super-resolution learning to text classification tasks and how they overcame them.

**Reproducibility:**

3: Could reproduce the results with some difficulty. The settings of parameters are underspecified or subjectively determined; the training/evaluation data are not widely available.

**Reviewer Confidence:**

3: Pretty sure, but there's a chance I missed something. Although I have a good feel for this area in general, I did not carefully check the paper's details, e.g., the math, experimental design, or novelty.

---

> ### Author Rebuttal · Authors · 2023-08-29
>
> We sincerely appreciate the time and effort you
> invested in reviewing our submission
>
> Regarding novelty, compared to existing approaches, ours does not generate any artificial data and relies on external high quality sources. Our results demonstrate  that mixing data with different resolutions works well, confirming that super-resolution, initially designed for image processing, is quite effective for text classification as well.
>
> We will improve the explanation of the methodology, pointing out its novelty. In addition, we will rewrite the algorithm to be clearer, illustrating it with an example.
>
>
> Answer Question A :
> High resolution-text and all other data-based data augmentation approaches we used follow the same technique: First, we use a given augmentation method (SR4NLP, generative, ...) to create a new set of data, then, when we split the tweets from the original dataset between training set and test set, we add our augmented data on the training set. Therefore we train on both the original data and the augmented data and only test on the original data.
>
> Answer Question B :
> This is indeed a misprint: there are 5 classes (Damage, Social, Advice-Warning, Other message, Not Useful).
>
> We would like to thank you again for your time and
> consideration.

---

### Meta-Review · Area_Chair_q4Jw · 2023-09-17

**Recommendation:** 3

**Metareview:**

This paper is about disaster urgency detection. It proposes a data augmentation method inspired by super resolution in image processing. The reviewers highlight the effectiveness of the proposed method (including for unseen disaster events). They also note that the application of super resolution to NLP classification tasks is new and interesting. The reviewers criticize some aspects of novelty, e.g., how the proposed method distinguishes from domain transfer techniques, which parts of the proposed approach are novel in NLP and what the specific challenges of this application were. Two reviewers note that the presentation and formalism requires revision. Finally, this work could benefit from further comparisons to better contextualize the reported results, e.g., by comparing against LLMs or other augmentation methods.

---

### Decision · Program_Chairs · 2023-10-07

**Decision:**

Accept-Findings

**Comment:**

This paper is about disaster urgency detection. It proposes a data augmentation method inspired by super resolution in image processing. The reviewers highlight the effectiveness of the proposed method (including for unseen disaster events). They also note that the application of super resolution to NLP classification tasks is new and interesting. The reviewers criticize some aspects of novelty, e.g., how the proposed method distinguishes from domain transfer techniques, which parts of the proposed approach are novel in NLP and what the specific challenges of this application were. Two reviewers note that the presentation and formalism requires revision. Finally, this work could benefit from further comparisons to better contextualize the reported results, e.g., by comparing against LLMs or other augmentation methods.